# Gallbladder adenocarcinomas undergo subclonal diversification and selection from precancerous lesions to metastatic tumors

**Minsu Kang**[1†], **Hee Young Na**[2†], **Soomin Ahn**[3*], **Ji-Won Kim**[1,4*], **Sejoon Lee**[5], **Soyeon Ahn**[6], **Ju Hyun Lee**[1], **Jeonghwan Youk**[1], **Haesook T Kim**[7], **Kui-Jin Kim**[8], **Koung Jin Suh**[1], **Jun Suh Lee**[9], **Se Hyun Kim**[1], **Jin Won Kim**[1], **Yu Jung Kim**[1], **Keun-Wook Lee**[1], **Yoo-Seok Yoon**[9], **Jee Hyun Kim**[1], **Jin-Haeng Chung**[2], **Ho-Seong Han**[9], **Jong Seok Lee**[1]

[1]Department of Internal Medicine, Seoul National University Bundang Hospital, Seongnam, Republic of Korea; [2]Department of Pathology, Seoul National University Bundang Hospital, Seongnam, Republic of Korea; [3]Department of Pathology and Translational Genomics, Samsung Medical Center, Sungkyunkwan University School of Medicine, Seoul, Republic of Korea; [4]Genealogy Inc, Seoul, Republic of Korea; [5]Center for Precision Medicine, Seoul National University Bundang Hospital, Seongnam, Republic of Korea; [6]Medical Research Collaboration Center, Seoul National University Bundang Hospital, Seongnam, Republic of Korea; [7]Department of Data Science, Dana Farber Cancer Institute, Harvard T.H. Chan School of Public Health, Boston, United States; [8]Biomedical Research Institute, Seoul National University Bundang Hospital, Seongnam, Republic of Korea; [9]Department of Surgery, Seoul National University Bundang Hospital, Seongnam, Republic of Korea

*For correspondence:
suminy317@gmail.com (SA);
jiwonkim@snubh.org (J-WK)

†These authors contributed equally to this work

Competing interest: The authors declare that no competing interests exist.

**Abstract** We aimed to elucidate the evolutionary trajectories of gallbladder adenocarcinoma (GBAC) using multi-regional and longitudinal tumor samples. Using whole-exome sequencing data, we constructed phylogenetic trees in each patient and analyzed mutational signatures. A total of 11 patients including 2 rapid autopsy cases were enrolled. The most frequently altered gene in primary tumors was *ERBB2* and *TP53* (54.5%), followed by *FBXW7* (27.3%). Most mutations in frequently altered genes in primary tumors were detectable in concurrent precancerous lesions (biliary intraepithelial neoplasia [BilIN]), but a substantial proportion was subclonal. Subclonal diversity was common in BilIN (n=4). However, among subclones in BilIN, a certain subclone commonly shrank in concurrent primary tumors. In addition, selected subclones underwent linear and branching evolution, maintaining subclonal diversity. Combined analysis with metastatic tumors (n=11) identified branching evolution in nine patients (81.8%). Of these, eight patients (88.9%) had a total of 11 subclones expanded at least sevenfold during metastasis. These subclones harbored putative metastasis-driving mutations in cancer-related genes such as *SMAD4*, *ROBO1*, and *DICER1*. In mutational signature analysis, six mutational signatures were identified: 1, 3, 7, 13, 22, and 24 (cosine similarity >0.9). Signatures 1 (age) and 13 (APOBEC) decreased during metastasis while signatures 22 (aristolochic acid) and 24 (aflatoxin) were relatively highlighted. Subclonal diversity arose early in precancerous lesions and clonal selection was a common event during malignant transformation in GBAC. However, selected cancer clones continued to evolve and thus maintained subclonal diversity in metastatic tumors.

## Editor's evaluation

This is the first dedicated study of clonal evolution in gallbladder cancer that involves precancerous, transformed primary and metastatic lesions. The main insights include the finding of subclonal diversity in precancerous lesions, a degree of bottlenecking during transformation but maintenance of some clonal complexity through metastases.

## Introduction

Gallbladder adenocarcinoma (GBAC) is a malignant neoplasm that has a high incidence rate in Chile, India, Poland, Pakistan, Japan, and Korea (*Roa et al., 2022*; *Valle et al., 2021*; *Bray et al., 2018*; *Bridgewater et al., 2014*). Surgery is currently the only curative treatment modality for GBAC. However, because most patients are diagnosed at an advanced stage and thus inoperable, they receive palliative chemotherapy only. Therefore, the prognosis is poor with a median overall survival of only 11–15 months (*Roa et al., 2022*; *Valle et al., 2021*; *Valle et al., 2010*).

The recent advancement of massively parallel sequencing technology has enabled us to deeply understand the genome of a variety of cancers. In GBAC, tumor suppressor genes such as *TP53*, *ARID1A,* and *SMAD4* and oncogenes such as *ERBB2* (*HER2*), *ERBB3*, and *PIK3CA* are significantly mutated (*Roa et al., 2022*; *Nepal et al., 2021*; *Wardell et al., 2018*; *Li et al., 2019*; *Nakamura et al., 2015*; *Lin et al., 2021*; *Narayan et al., 2019*; *Li et al., 2014*). Of note, *ERBB2* amplification and overexpression occur in approximately 6.9–28.6% of GBAC (*Roa et al., 2022*; *Narayan et al., 2019*; *Nakazawa et al., 2005*; *Nam et al., 2016*) and may have therapeutic implications.

Cancer cells undergo clonal evolution by acquiring additional mutations and thus exhibit more aggressive phenotypes, including invasion and metastasis (*Turajlic et al., 2019*; *Greaves and Maley, 2012*; *Davis et al., 2017*). Several large-scale studies have provided evidence of clonal evolution in some cancer types, including lung and kidney cancers (*Jamal-Hanjani et al., 2017*; *Turajlic et al., 2018*). However, no study has analyzed the patterns of clonal evolution from the initiation of carcinogenesis to distant metastasis in patients with GBAC.

This study aims to analyze the clonal evolutionary trajectories during carcinogenesis and metastasis of GBAC using multi-regional and longitudinal specimens including precancerous lesions (biliary intraepithelial neoplasia [BilIN]), primary tumors, and metastatic tumors from patients who underwent biopsy, surgery, and rapid autopsy.

**Table 1.** Baseline characteristics of 11 patients with GBAC.

| Patient ID* | Sex | Age at diagnosis | ECOG PS at diagnosis | Stage at diagnosis | Differentiation |
|---|---|---|---|---|---|
| GB-A1 | F | 70 | 1 | IV | PD |
| GB-S1 | F | 66 | 1 | IV | MD |
| GB-S2 | F | 66 | 1 | IV | MD |
| GB-S3 | F | 75 | 1 | III | MD |
| GB-A2 | M | 61 | 1 | IV | MD |
| GB-S4 | M | 72 | 1 | IV | MD |
| GB-S5 | F | 70 | 1 | IV | PD |
| GB-S6 | M | 70 | 0 | IV | MD |
| GB-S7 | F | 74 | 1 | IV | PD |
| GB-S8 | M | 67 | 1 | III | WD |
| GB-S9 | M | 59 | 0 | IV | MD |

*In patient ID, 'A' indicates autopsy cases whereas 'S' indicates surgery cases.

GBAC = gallbladder adenocarcinoma. ECOG PS = Eastern Cooperative Oncology Group performance status. M = male. F = female. PD = poorly differentiated. MD = moderately differentiated. WD = well-differentiated.

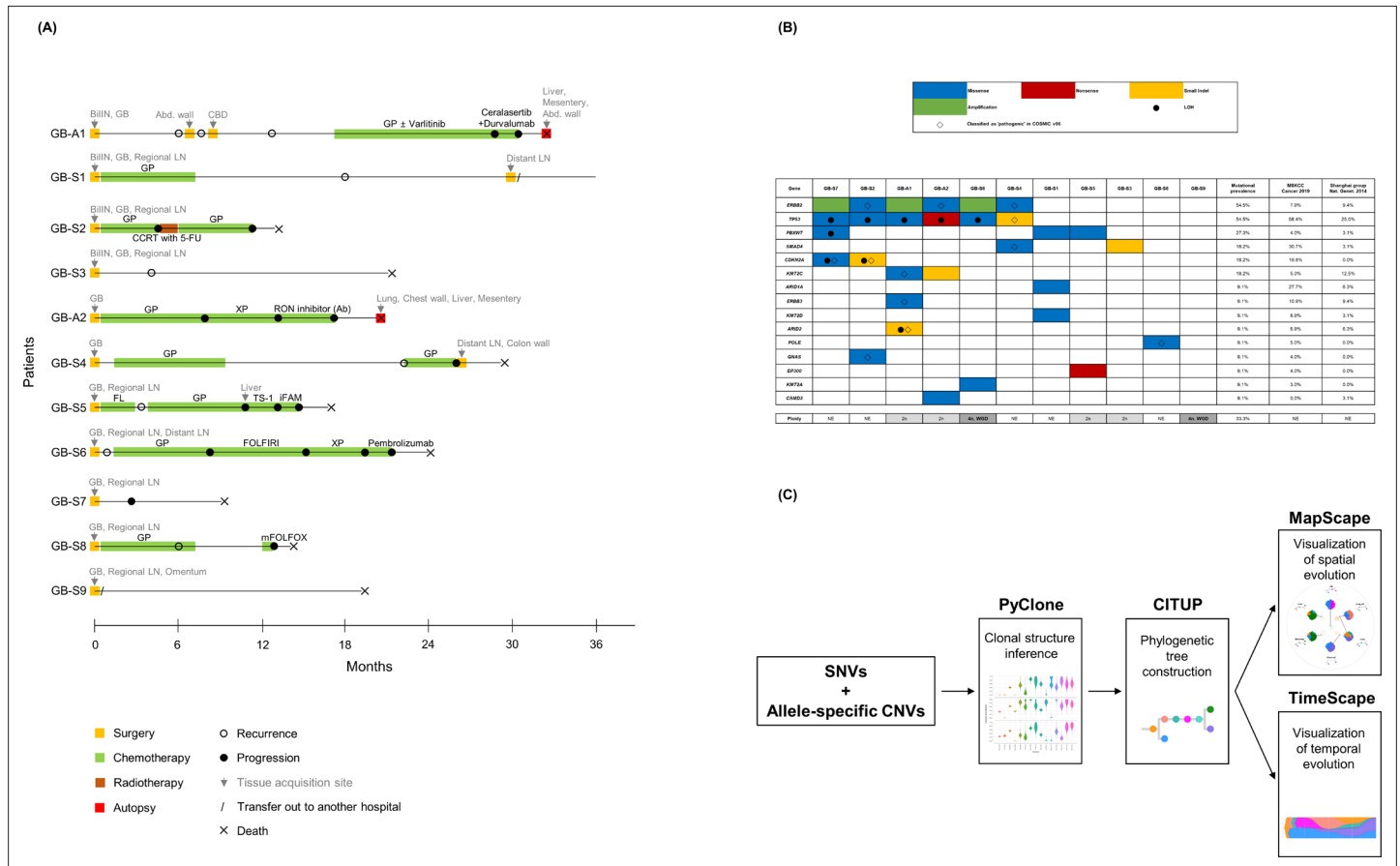

**Figure 1.** Clinical history of patients, mutational landscape, and study workflow. (**A**) Clinical history of 11 patients is summarized in the swimmer plot. (**B**) The mutational landscape of 11 primary tumors (GB) is visualized and compared with the two previous studies on gallbladder adenocarcinoma (GBAC). (**C**) The process of constructing clonal evolution trajectories using multiple tumor samples is shown in the workflow. Ab, antibody; Abd, abdominal; Adj, adjuvant; BilIN, biliary intraepithelial neoplasia; CBD, common bile duct; CCRT, concurrent chemoradiotherapy; CNVs, copy number variations; COSMIC, Catalogue Of Somatic Mutations In Cancer; FL, 5-fluorouracil + leucovorin; FOLFIRI, 5-fluorouracil + leucovorin + irinotecan; GB, gallbladder; GP, gemcitabine + cisplatin; iFAM, infusional 5-fluorouracil + doxorubicin + mitomycin-C; LOH, loss of heterozygosity; mFOLFOX, modified 5-fluorouracil + leucovorin + oxaliplatin; MSKCC, Memorial Sloan Kettering Cancer Center; NE, not evaluable; SNVs, single nucleotide variants; WGD, whole genome doubling; XP, capecitabine + cisplatin; 5-FU, 5-fluorouracil.

The online version of this article includes the following figure supplement(s) for figure 1:

**Figure supplement 1.** Inference of clonal structure by PyClone algorithm.

**Figure supplement 2.** BilIN and primary gallbladder adenocarcinoma (GBAC) of the GB-S7 patient presumed to be derived from different origins.

# Results

## Baseline characteristics of patients with GBAC

A total of 11 patients, including 2 rapid autopsy cases (GB-A1 and GB-A2) and 9 surgery cases (GB-S1 – 9), were enrolled in this study (*Table 1* and *Figure 1A*). There were 5 male and 6 female patients with a median age was 70 years (range, 59–75 years). Two patients had stage III and nine patients had IV disease at diagnosis. A total of 58 samples were analyzed, including 11 pairs of matched primary tumors and normal tissues, 6 concurrent BilIN, and 30 metastatic tumors. Among them, 15 samples obtained by rapid autopsy were fresh-frozen, and 43 samples obtained by surgery or biopsy were formalin-fixed paraffin-embedded (FFPE) (*Supplementary file 1A*). The median number of filtered somatic single nucleotide variants (SNVs) and small indels was 61 (range, 12–241). The number of metastatic tumors in each patient ranged from 1 to 11.

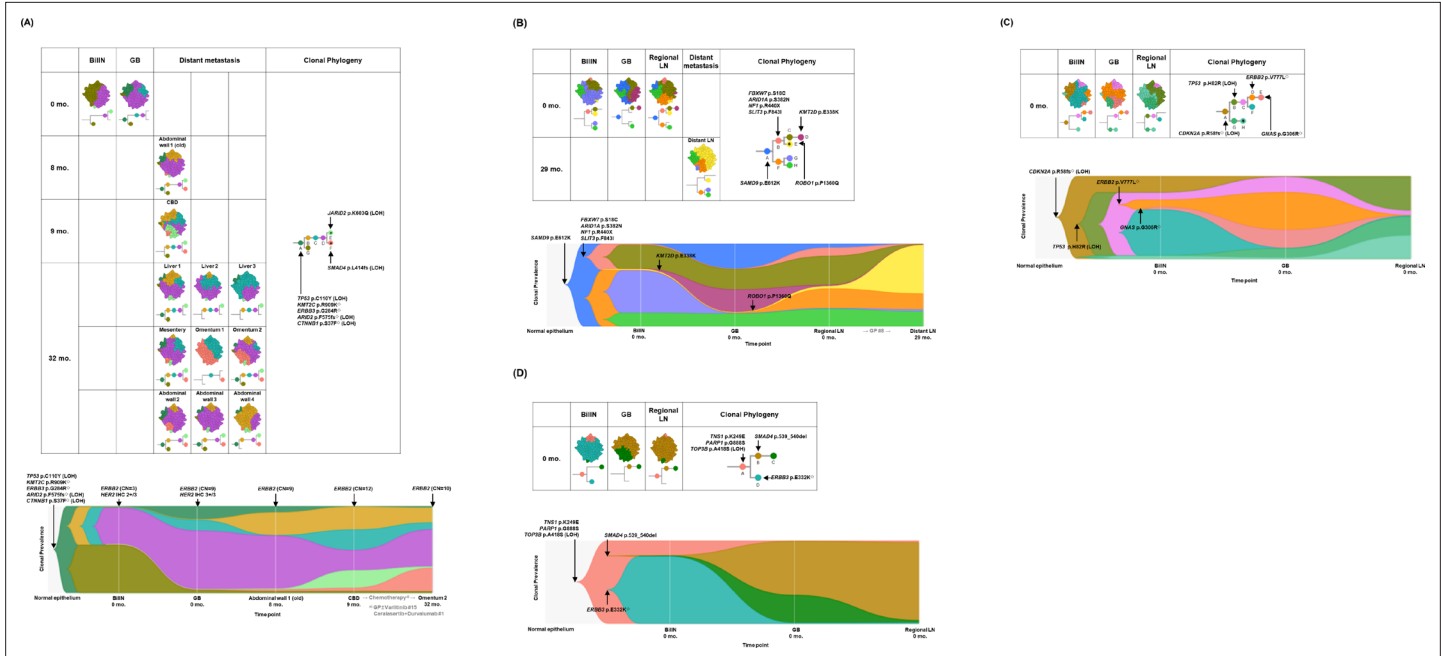

**Figure 2.** Spatial and temporal clonal evolution of four patients with gallbladder adenocarcinoma (GBAC) whose precancerous BilIN tissues were analyzed. (**A–D**) The most probable phylogenetic trees and MapScape and TimeScape results are visualized for GB-A1 (**A**), GB-S1 (**B**), GB-S2 (**C**), and GB-S3 (**D**). In MapScape visualization (table), the numbers in the row indicate the time taken from diagnosis to tissue acquisition in months, and the columns indicate whether the tissue is a precancerous lesion, primary tumor, or metastatic lesion. Colors represent distinct clones and clonal prevalences per site were proportional to the corresponding-colored area of the cellular aggregate representation. In TimeScape visualization (schema), clonal prevalences (vertical axis) were plotted across time points (horizontal axis) for each clone (colors). Asterisks (*) in the clonal phylogenetic tree denote subclones that constituted <5% in the primary tumor and expanded more than sevenfold in the metastatic tumor. Diamond symbol (◇) is used if the mutation in the figure is classified as 'pathogenic' in Catalogue Of Somatic Mutations In Cancer (COSMIC) v96. Notable events were marked with arrows. The time from diagnosis of GBAC to tissue acquisition was indicated under the sample name. Chemotherapy history was indicated in gray color, where '#' represents the number of chemotherapy cycles. BilIN, biliary intraepithelial neoplasia; CBD, common bile duct; CN, copy number; GB, gallbladder; GP, gemcitabine + cisplatin; LN, lymph node; LOH, loss of heterozygosity.

## Mutational landscape and ploidy of primary tumors

The mutational landscape of the 11 primary tumors (GB) was analyzed and compared with previous literature (*Figure 1B*; *Narayan et al., 2019*; *Li et al., 2014*). The most frequently altered gene in the primary tumor was *ERBB2* and *TP53* (54.5%), followed by *FBXW7* (27.3%). Of the six *ERBB2* alterations, three were amplification and the other three were missense mutations classified as 'pathogenic' in the COSMIC (Catalogue Of Somatic Mutations In Cancer) v96 database. Among six *TP53* mutations found in six patients, five were accompanied by loss of heterozygosity (LOH).

Ploidy was analyzed in 15 tumors of six patients with purity >0.4 because ploidy estimation was inaccurate when tumor purity is ≤0.4 (*Favero et al., 2015*). In two patients (GB-S6 and GB-S9, 33.3%), whole genome doubling (WGD) was detected in both the primary and metastatic tumors (*Figure 1B*). In GB-S5 patient, WGD was found in distant metastasis, but not in primary GBAC.

## Somatic mutations developed at the precancerous stage

Multi-regional distribution and longitudinal evolution of clones were analyzed using PyClone (*Figure 1—figure supplement 1* and *Supplementary file 2*; *Roth et al., 2014*) and CITUP (*Malikic et al., 2015*) and then visualized using MapScape and TimeScape (*Smith et al., 2017*; *Figure 1C*). Among six patients having concurrent BilIN tissues, two patients were excluded from the further analysis. One patient had low tumor purity of BilIN (0.03) and the other patient had different truncal mutations of BilIN and primary GBAC, suggesting different origins of the two tumors (*Figure 1—figure supplement 2*).

Most mutations in frequently altered genes in GBAC existed at the BilIN stage (10 of 13, 76.9%), but some of them were subclonal. In GB-A1 (*Figure 2A*), *TP53* C100Y with LOH, *KMT2C* R909K,

*ERBB3* G284R, *ARID2* F575fs with LOH, and *CTNNB1* S37F with LOH were observed clonally in BilIN. In GB-S1 (**Figure 2B**), *SAMD9* E612K was clonal whereas *FBXW7* S18C, *ARID1A* S382N, and *NF1* R440X were subclonal. In GB-S2 (**Figure 2C**), considering the cellular prevalence of mutations, it is speculated that the mutations developed in the order of *CDKN2A* R58fs with LOH, *TP53* H82R with LOH, *ERBB2* V777L, and *GNAS* G306R during carcinogenesis. In GB-S3 (**Figure 2D**), *ERBB3* E332K was dominant in BilIN, while *SMAD4* 539_540del was not detected.

## Subclonal diversity and 'selective sweep' phenomenon during the early stage of carcinogenesis

Branching evolution and subclonal diversity were commonly observed in the BilIN of the four patients (**Figure 2**). When compared with the concurrent primary tumors, one subclone commonly shrank in the primary tumors, while the other subclones that acquired additional mutations relatively expanded in the primary tumors, suggesting a selective sweep phenomenon (**Merlo et al., 2006**). Selected subclones underwent linear and branching evolution, and thus subclonal diversity was maintained after the BilIN stage. In GB-A1 (**Figure 2A**), clone A underwent branching evolution into B and G, and clone B linearly evolved into C and then D. Clone D, which acquired additional mutations, increased from 42.7% to 68.1% while clone G decreased from 55.7% to 2.8%. In GB-S1 (**Figure 2B**), clone D that acquired *KMT2D* E338K increased from 0.1% to 28.1%, while clone G decreased from 51.4% to 0%. In GB-S2 (**Figure 2C**), clone D acquired *ERBB2* V777L, clone E acquired *GNAS* G306R, and clone G increased from 9.0%, 3.5%, and 0% to 45.5%, 22.3%, and 11.4%, respectively. In contrast, clone F decreased from 55.8% to 0.6%. In GB-S3 (**Figure 2D**), clone B that acquired *SMAD4* 539_540del increased from 0.1% to 65.6%, while clone D containing *ERBB3* E332K decreased from 81.1% to 0.6%.

**Table 2.** List of subclones expanding during metastasis.

| Patient ID | Subclone | No. of mutations | Putative driver mutations | Clonal prevalence in primary tumor | Clonal prevalence in metastasis | Metastatic organ |
|---|---|---|---|---|---|---|
| GB-A1 | E | 121 | *JARID2* p.K603Q (LOH) | 0.2% | 20.5% | CBD |
| | F | 54 | *SMAD4* p.L414fs (LOH) | 0.4% | 49.8% | Omentum 1 |
| | | | | | 27.0% | Omentum 2 |
| GB-S1 | E | 6 | *ROBO1* p.P1360Q | 0.0% | 59.2% | Distant LN |
| GB-S2 | H | 6 | – | 1.2% | 28.1% | Regional LN |
| GB-A2 | F | 14 | *PRKCD* p.I153L | 0.3% | 73.1% | Liver |
| | | | | | 68.3% | Mesentery |
| | G | 3 | *DICER1* p.T519A | 2.8% | 39.9% | Lung |
| | | | | | 61.5% | Chest wall |
| GB-S4 | F | 32 | *FBXW2* p.W450C | 0.0% | 34.1% | Colon wall |
| | H | 36 | – | 0.0% | 34.9% | Distant LN |
| GB-S5 | B | 28 | *KIAA0100* p.F5S | 1.2% | 54.4% | Liver |
| | | | *CSMD2* p.E411K | | | |
| GB-S6 | B | 7 | *OSCP1* p.R351X | 1.1% | 60.2% | Lung |
| GB-S7 | C | 5 | – | 3.3% | 24.7% | Regional LN |

Putative driver mutations are indicated, and a full list of mutated genes is specified in **Supplementary file 2**.
GB = gallbladder. LN = lymph node. CBD = common bile duct.

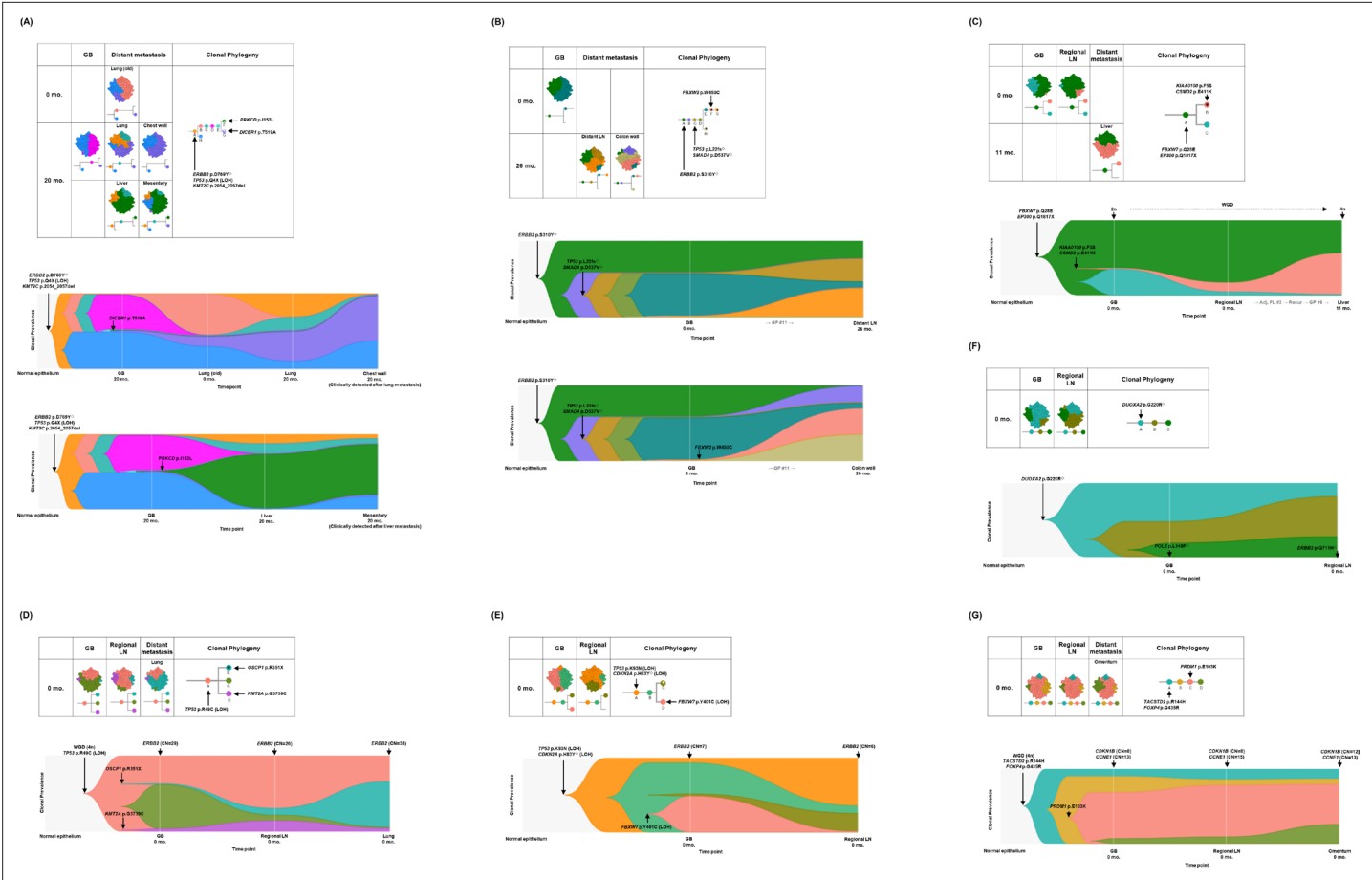

**Figure 3.** Spatial and temporal clonal evolution of additional seven patients with gallbladder adenocarcinoma (GBAC). (**A–G**) The most probable phylogenetic trees and MapScape and TimeScape results are visualized for GB-A2 (**A**), GB-S4 (**B**), GB-S5 (**C**), GB-S6 (**D**), GB-S7 (**E**), GB-S8 (**F**), and GB-S9 (**G**). In MapScape visualization (table), the numbers in the row indicate the time taken from diagnosis to tissue acquisition in months, and the columns indicate whether the tissue is a precancerous lesion, primary tumor, or metastatic lesion. Colors represent distinct clones and clonal prevalences per site were proportional to the corresponding-colored area of the cellular aggregate representation. In TimeScape visualization (schema), clonal prevalences (vertical axis) were plotted across time points (horizontal axis) for each clone (colors). Asterisks (*) in the clonal phylogenetic tree denote subclones that constituted <5% in the primary tumor and expanded more than sevenfold in the metastatic tumor. Diamond symbol (◊) is used if the mutation in the figure is classified as 'pathogenic' in Catalogue Of Somatic Mutations In Cancer (COSMIC) v96. Notable events were marked with arrows. The time from diagnosis of GBAC to tissue acquisition was indicated under the sample name. Chemotherapy history was indicated in gray color, where '#' represents the number of chemotherapy cycles. Adj, adjuvant; BilIN, biliary intraepithelial neoplasia; CN, copy number; FL, 5-fluorouracil + leucovorin; GB, gallbladder; GP, gemcitabine + cisplatin; LN, lymph node; LOH, loss of heterozygosity; WGD, whole genome doubling.

## Evolutionary trajectories and expansion of subclones during regional and distant metastasis

Combined analysis of regional and distant metastatic tumors revealed branching and linear evolution in nine patients (81.8%) and linear evolution only in two patients (18.2%). Of the nine patients with branching and linear evolution, eight (88.9%) had a total of 11 subclones expanded at least sevenfold in the regional or distant metastasis stage (*Table 2*). In GB-A1 (*Figure 2A*), clone E, which acquired *JARID2* (*Celik et al., 2018*) K603Q with LOH, increased from 0.2% to 20.5% during common bile duct metastasis. In addition, clone F, which acquired *SMAD4* (*Narayan et al., 2019*; *Jamal-Hanjani et al., 2017*; *Yoon et al., 2021*; *Zhao et al., 2018*) L414fs with LOH, expanded from 0.4% to 49.8% during omentum 1 metastasis and to 27.0% during omentum 2 metastasis. In GB-S1 (*Figure 2B*), clone B harboring *SLIT3* F843I mutation evolved into E by additionally acquiring *ROBO1* P1360Q mutation (*Gara et al., 2015*). In GB-A2 (*Figure 3A*), clone F, which acquired *PRKCD* (*Griner and Kazanietz, 2007*; *Yoshida, 2007*) I153L, expanded from 0.3% to 73.1% and 68.3% during metastasis to liver and mesentery, respectively. In addition, clone G acquired *DICER1* (*Foulkes et al., 2014*) T519A and

expanded from 2.8% to 39.9% and 61.5% during metastasis to the lung and chest wall, respectively. In the other five patients (*Figures 2C and 3B–E*), mutations in *FBXW2* (*Xu et al., 2017*), *KIAA0100* (*Gara et al., 2018*), *CSMD2* (*Gara et al., 2018*), and *OSCP1* (*Yi et al., 2017*) genes were observed.

In GB-S8 and GB-S9, only linear evolution was identified (*Figure 3F and G*). In GB-S9, amplification of cell cycle-related oncogenes *CDKN1B* and *CCNE1* was uniformly observed from primary GBAC to regional and distant metastatic tumors.

## Polyclonal metastasis and intermetastatic heterogeneity

The metastatic lesions were uniformly polyclonal. In GB-A1, GB-A2, and GB-S4, which contained two or more distant metastatic lesions, the clonal composition of tissues obtained from the same or adjacent organs showed a similar tendency, while the clonal composition of anatomically distant organs was distinct from each other. In GB-A1 (*Figure 2A*), abdominal wall 1–4 did not contain clone C, and liver 1–3 did not contain clone F. In addition, omentum 1–2 had a high prevalence of clone F of over 27.0%. In GB-A2 (*Figure 3A*), as opposed to clone G, which was more prevalent in lung and chest wall lesions, clone F was more prevalent in liver and mesentery lesions. In clinical information of the GB-A2 patient, chest wall and mesentery metastases developed later than lung and liver metastases. In GB-S4 (*Figure 3B*), proportions of clones G and H were specifically high in distant LN and colon wall metastasis, respectively.

## Mutational signatures during clonal evolution

We identified six mutational signatures (cosine similarity >0.9): signatures 1 (age), 3 (DNA double-strand break-repair), 7 (ultraviolet), 13 (APOBEC), 22 (aristolochic acid), and 24 (aflatoxin). Then, our results were compared with those of the Memorial Sloan Kettering Cancer Center (MSKCC) and Shanghai datasets (*Narayan et al., 2019*; *Li et al., 2014*). While MSKCC and Shanghai datasets consist merely of primary tumors, our dataset includes precancerous and metastatic lesions. Signatures 1, 3, and 13 were commonly dominant in all three datasets, while signatures 22 and 24 were exclusive in our dataset (*Figure 4A*). In our dataset, the limited number of SNVs and small indels per sample (median 61, range 12–241) made it difficult to compare among individual tumors (*Figure 4—figure supplement 1*). Therefore, we classified the mutations according to the timing of development during clonal evolution (*Figure 4B*): (1) early carcinogenesis (i.e., clone A in *Figures 2 and 3*), (2) late carcinogenesis (i.e., subclones which were not categorized in early carcinogenesis or metastasis), and (3) metastasis (i.e., subclones expanding during metastasis [*Table 2*]). At the metastasis phase, signatures 1 and 13 decreased while signatures 22 and 24 increased compared with early and late carcinogenesis. Then, the same analysis was performed according to the type of sample (*Figure 4—figure supplement 2*): (1) BilIN, (2) GB, (3) regional LN metastasis, and (4) distant metastasis. This mutational signature analysis was conducted using the published tool Mutalisk (*Lee et al., 2018*) and then validated with two additional tools, Signal (*Degasperi et al., 2020*) and MuSiCa (*Díaz-Gay et al., 2018*; *Supplementary file 1B* and *Figure 4—figure supplement 3*).

## *ERBB2* amplification during clonal evolution

In GB-A1, *ERBB2* copy number gain might have initiated from the precancerous stage and further progressed during the malignant transformation of BilIN. *ERBB2* copy number (*Figure 5A and B*) was 3 and 9 in concurrent BilIN and primary tumors, respectively. The increased copy number of *ERBB2* was maintained after distant metastasis (*Figure 2A*). HER2 (=*ERBB2*) silver in situ hybridization (SISH) (*Figure 5C and D*) was conducted and *ERBB2* copy number per cell ranged from 1 to 5 in BilIN, and from 2 to 14 in primary GBAC. *ERBB2*/CEP17 ratio was 2.48 and 6.00 in BilIN and primary GBAC, respectively (*Figure 5E*). In addition, HER2 immunohistochemistry (IHC) (*Figure 5F and G*) was carried out to evaluate whether the *ERBB2* amplification was correlated to HER2 protein expression levels on the membrane of tumor cells. The HER2 IHC results were 2+/3 in the BilIN and 3+/3 in primary GBAC. In GB-S6 and GB-S7 (*Figure 3D and E*), *ERBB2* amplification was uniformly identified from primary tumors to metastatic tumors.

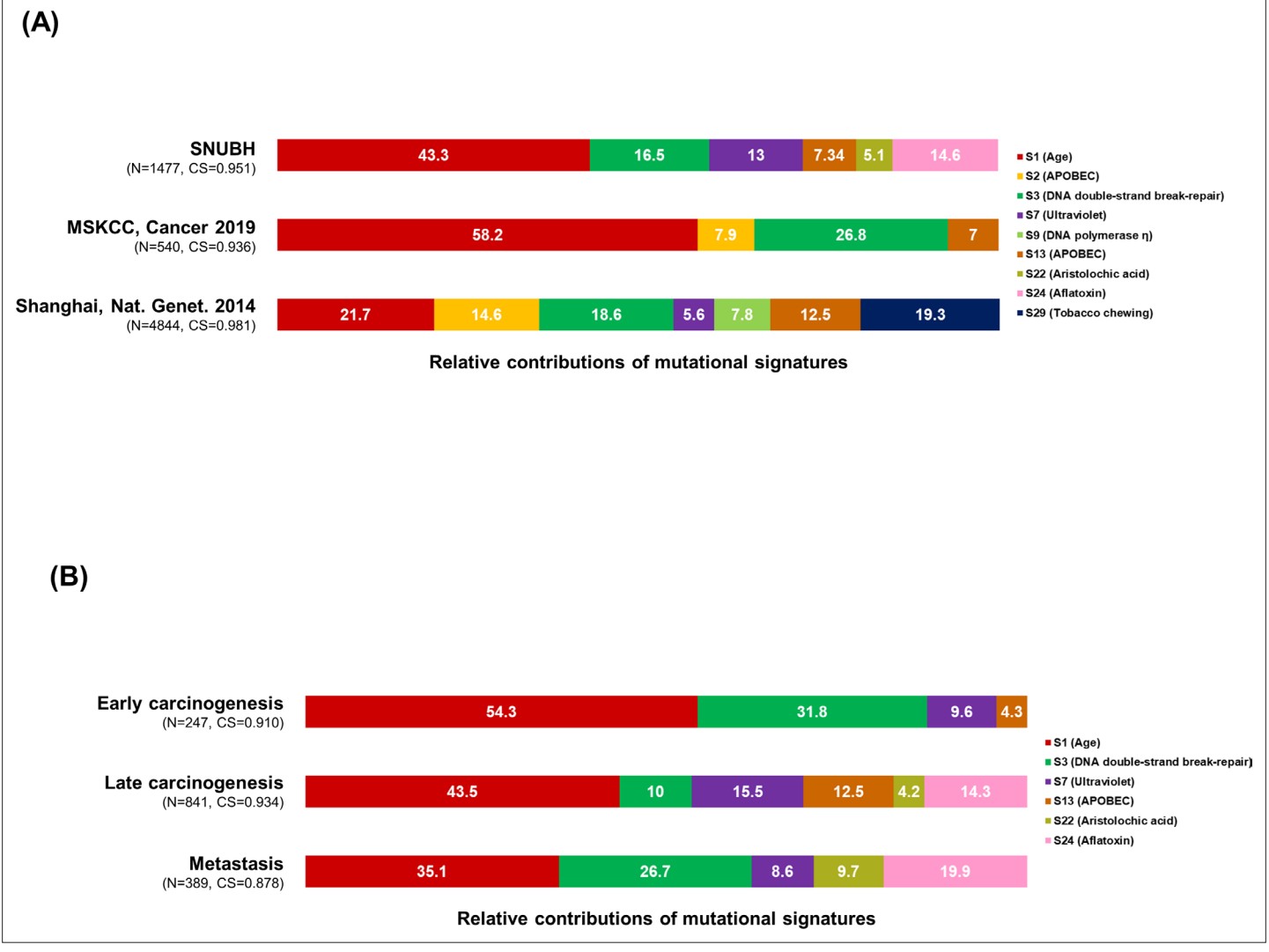

**Figure 4.** Mutational signature analysis. (**A–B**) The 100% stacked bar plots compare the proportions of known COSMIC Mutational Signatures v2 within our dataset and two public (MSKCC and Shanhai) datasets (**A**), and each category split according to the timing of development during clonal evolution (**B**). The total number of mutations (N) and cosine similarity (CS) values of each category were noted. BilIN, biliary intraepithelial neoplasia; COSMIC, Catalogue Of Somatic Mutations In Cancer; GB, gallbladder; LN, lymph node; MSKCC, Memorial Sloan Kettering Cancer Center.

The online version of this article includes the following figure supplement(s) for figure 4:

**Figure supplement 1.** Mutational signatures within each sample from 11 gallbladder adenocarcinoma (GBAC) patients.

**Figure supplement 2.** Mutational signatures within each category split according to the type of sample.

**Figure supplement 3.** Mutational signature analysis validated by two additional tools, Signal and MuSiCa.

## Discussion

To the best of our knowledge, this is the first study to investigate clonal evolution from precancerous lesions to metastatic tumors in patients with GBAC. In this study, evolutionary trajectories of GBAC were inferred using multi-regional and longitudinal whole-exome sequencing (WES) data from precancerous lesions to primary and metastatic tumors. Based on these results, we derived comprehensive models of carcinogenesis and metastasis in GBAC.

In our analysis of carcinogenesis, we discovered three common themes. First, most mutations in frequently altered genes in primary GBAC are detected in concurrent BilIN (10 of 13, 76.9%), but a substantial proportion was subclonal. Second, branching evolution and subclonal diversity are commonly observed at the BilIN stage. Third, one subclone in BilIN commonly shrinks in the primary tumors, while the other subclones undergo linear and branching evolution, maintaining subclonal

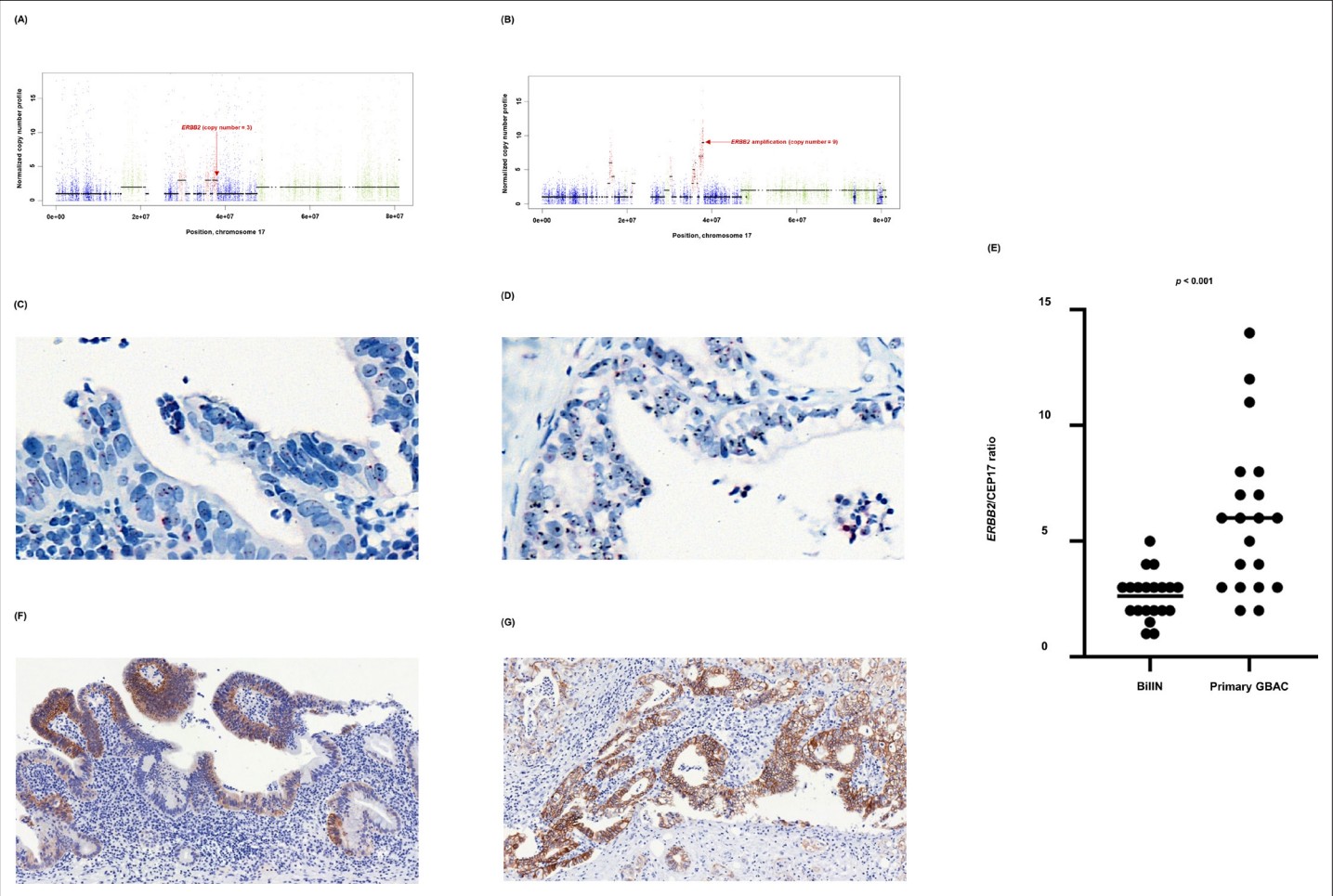

**Figure 5.** *ERBB2* copy number variation during neoplastic transformation of BilIN in GB-A1. (**A–G**) *ERBB2* gene amplification (**A and B**), *HER2* SISH (**C and D**), and *HER2* IHC (**F and G**) were compared between BilIN (**A, C**) and (**F**) and primary GBAC (**B**), (**D**), and (**G**) samples and the mean *ERBB2*/CEP17 ratio of BilIN and GB-A1 samples were compared by using the Wilcoxon rank-sum test (**E**). BilIN, biliary intraepithelial neoplasia; GBAC, gallbladder adenocarcinoma; IHC, immunohistochemistry; SISH, silver in situ hybridization.

diversity after the BilIN stage. A previous study in colorectal cancer by Vogelstein and colleagues demonstrated a stepwise carcinogenesis model from the precancerous lesion, adenoma, to invasive carcinoma by the accumulation of mutations, called the adenoma-carcinoma sequence (*Vogelstein et al., 1988*). In addition, recent studies on esophageal squamous cell carcinoma have reported that not only dysplasia but also histologically normal epithelia frequently harbor cancer-driving mutations (*Chen et al., 2017*; *Yokoyama et al., 2019*).

A recent study on GBAC (*Lin et al., 2021*) reported that *CTNNB1* mutation was frequently observed (5 out of 11) when BilIN and primary GBAC coexist. In addition, *ERBB2* alteration was found in 36.4%. In our study, *CTNNB1* S37F with LOH was observed in one (GB-A1) of four patients and 54.5% of patients had an *ERBB2* alteration. On the other hand, two previous studies on GBAC (*Narayan et al., 2019*; *Li et al., 2014*) reported the prevalence of *ERBB2* alteration as low as 7.9% and 9.4%, respectively. Race or lifestyle differences might have contributed to the difference.

In the analysis of metastasis, the following three phenomena are observed. First, subclonal expansion is frequent (8 of 11 patients, 72.7%) and some subclones expand substantially in metastatic tumors, leading to increased subclonal diversity. Previous studies suggest that subclonal diversity increases through branching evolution during progression and metastasis (*Turajlic and Swanton, 2016*; *Hong et al., 2015*; *Yachida et al., 2010*; *Minussi et al., 2021*). Second, metastases are polyclonal but metastatic lesions in one organ or adjacent organs show similar clonal compositions. Previously, it was thought that metastasis is initiated by the migration of a single cell

to another organ (*Nowell, 1976*). However, recent data suggest that polyclonal seeding occurs due to the migration of a cluster of cancer cells (*Turajlic et al., 2019*; *Cheung and Ewald, 2016*; *Ullah et al., 2018*; *Wei et al., 2017*). Third, we found the possibility of metastasis to metastasis spread. In recent studies on prostate and breast cancers, metastasis-to-metastasis spread was frequent (*Ullah et al., 2018*; *Gundem et al., 2015*). Our study found the possibility of metastasis-to-metastasis spread in GB-A2 (*Figure 3A*). Although intratumoral heterogeneity of primary GBAC may make it difficult to draw a strong conclusion, our data may support the possibility of metastasis-to-metastasis spread.

Of the 11 expanded subclones at the metastasis stage, we described putative driver mutations in eight subclones based on the previous literature (*Table 2*). For example, *SMAD4* mutations expanded during metastasis in GB-A1 and GB-S3 (*Figure 2A and D*) and have been associated with distant metastasis and poor prognosis in various cancers, including GBAC (*Narayan et al., 2019*; *Jamal-Hanjani et al., 2017*; *Yoon et al., 2021*; *Zhao et al., 2018*). In GB-S1 (*Figure 2B*), clone B containing *SLIT3* F843I evolves into E during metastasis by acquiring *ROBO1* P1360Q. The SLIT/ROBO pathway suppresses tumor progression by regulating invasion, migration, and apoptosis (*Gara et al., 2015*). In GB-A2 (*Figure 3A*), *DICER1* T519A is found in lung and chest wall metastases. The *DICER1* gene is associated with pleuropulmonary blastoma in children (*Foulkes et al., 2014*). In GB-S5 (*Figure 3C*), *KIAA0100* F5S and *CSMD2* E411K are found during metastasis. *KIAA0100* and *CSMD2* are frequently mutated during metastasis in adrenocortical carcinoma (*Gara et al., 2018*).

In mutational signature analysis, considering the evolutionary trajectories in cancers, we suggest that the timing of development during clonal evolution (*Figure 4B*) is better classification criterion than the type of sample (*Figure 4—figure supplement 2*). Our data indicate that the importance of signatures 1 and 13 decreased during metastasis while the roles of signatures 22 and 24 were relatively highlighted. Aristolochic acid is an ingredient of oriental herbal medicine (*Debelle et al., 2008*; *Hoang et al., 2013*). In addition, aflatoxin is known to be contained in soybean paste and soy sauce (*Ok et al., 2007*). Taken together, the two carcinogens might have little impact on the early stage of cancer development, but their impacts might be highlighted in overt cancer cells.

In this cancer precision medicine era, targeted sequencing data of a single specimen are not enough to determine whether the detected mutations are clonal or subclonal. This proof-of-concept study may enable us to deeply understand the clonal evolution in GBAC. Moreover, we found that some of the mutations were clonal while a substantial proportion was subclonal, which is usually not an effective druggable target. For example, if a drug targeting *ERBB2* p.V777L, a pathogenic mutation, is administered to GB-S2 patient (*Figure 2C*), the therapeutic effect will be limited in subclones without the *ERBB2* p.V777L mutation, especially from regional metastasis. Therefore, we believe that our study highlights the importance of precise genomic analysis of multi-regional and longitudinal samples in individual cancer patients. However, one caveat is that we cannot easily apply this to real-world patients because multi-regional and longitudinal tumor biopsies may not be feasible in most patients unless they underwent surgery and repeated biopsies. Recent studies using circulating tumor DNA have shown the possibility of easily detecting mutations involved in cancer development and progression (*Ignatiadis et al., 2021*). By detecting clonal mutations from the carcinogenesis stage in healthy individuals, we can diagnose GBAC at an early stage. In addition, by detecting subclonal mutations in patients with GBAC, we can monitor the expanding subclones during follow-up, which enables us to detect cancer progression earlier.

This study has several limitations. First, it is not possible to obtain samples through frequent biopsies whenever desired. Thus, tumor samples were not acquired according to their developmental sequence. Second, due to intratumoral heterogeneity, the clonal composition of a small piece of the tumor may not reflect that of the entire lesion (*Marusyk et al., 2012*). Third, as the number of analyzed samples was different among patients, an accessibility bias was inevitable – the more samples the patient has, the more clones we can identify.

In conclusion, subclonal diversity developed early in precancerous lesions and clonal selection was a common event during malignant transformation in GBAC. However, cancer clones continued to evolve in metastatic tumors and thus maintained subclonal diversity. Our novel approach may help us to understand the GBAC of individual patients and to move forward to precision medicine that enables early detection of carcinogenesis and metastasis, and effective targeted therapy in these patients.

## Methods

### Patients and tumor samples

Patients were eligible for this study if they were diagnosed with GBAC and received surgery between 2013 and 2018 at Seoul National University Bundang Hospital (SNUBH), Seongnam, Korea. Of the 272 patients who underwent surgery, 9 patients who were able to obtain ≥3 types of tissue from among BilIN, GBAC, regional LN metastasis, and distant metastasis and died at the time of analysis were enrolled in the study. In addition, two patients who donated their bodies for rapid autopsy were included in the study. Our board-certified pathologists (Prof. Soomin Ahn and Hee Young Na) identified and dissected BilIN and GBAC lesions from tissue slides. Patients' clinical information was obtained through retrospective medical record reviews. This study was approved by the Institutional Review Board (IRB) of SNUBH (IRB No. B-1902/522-303). Informed consent was waived because of the retrospective and anonymous nature of the study.

### WES of GBAC

DNA was extracted from fresh-frozen or FFPE tissues. Library preparation and exome capture was carried out using Agilent SureSelect^XT Human All Exon V6 (Santa Clara, CA, USA). WES was conducted with a paired-end, 100 bp using Illumina NovaSeq 6000 (San Diego, CA, USA). The depth of coverage of tumors and normal control samples were at least 300× and 200×, respectively.

### Analysis of WES data

WES data of GBAC and matched normal samples were analyzed using the Genomon2 pipeline (Institute of Medical Science, University of Tokyo, Tokyo, Japan; https://genomon.readthedocs.io/ja/latest/, accessed on February 1, 2022). In brief, sequencing reads from adapter-trimmed.fastq files were aligned to the human reference genome GRCh37 (hg19) without the 'chr' prefix using Burrows-Wheeler Aligner version 0.7.12, with default settings. SNVs and small indels were called by eliminating polymorphisms and sequencing errors and filtered by pre-specified criteria used in previous literature analyzed with the Genomon2 pipeline (*Yokoyama et al., 2019*; *Kakiuchi et al., 2020*; *Ochi et al., 2021*): (a) only exonic or splicing sites were included; (b) synonymous SNVs, unknown variants, or those without proper annotation were excluded; (c) polymorphisms in dbSNP 131 were excluded; (d) p-values <0.01 from Fisher's exact test were included; (e) simple repeat sequences were excluded; (f) strand ratio between positive-strand and negative-strand should not be 0 or 1 in tumor samples; (g) the number of variant reads should exceed 4 in tumor samples. For each patient, filtered variant lists of tumor samples were merged. Then, the merged list of target variants was manually called in each. bam file using bam-readcount version 0.8.0 (https://github.com/genome/bam-readcount; *Khanna et al., 2022a*; *Khanna et al., 2022b*, accessed on February 1, 2022) with Phred score and mapping quality of more than 30 and 60, respectively.

For samples with tumor purity >0.4, the ploidy of tumor cells was estimated using Sequenza version 3.0.0 to identify WGD (*Favero et al., 2015*). Copy number variations were analyzed using the Control-FREEC version 11.5 (*Boeva et al., 2012*). Gene amplification was defined as a copy number ≥6 (*Wolff et al., 2018*). In brief, the aligned.bam files were converted to.pileup.gz format using SAMtools version 1.9 (*Li, 2011*). The .pileup.gz files of data from GBAC and matched normal samples were analyzed using Control-FREEC with default settings. These datasets were then used to statistically infer clonal population structure using PyClone version 0.13.0 with the 'pyclone_beta_binomial' model (*Roth et al., 2014*). Clonal phylogeny was inferred by CITUP version 0.1.0 (*Malikic et al., 2015*) using cellular prevalence values of each cluster which were generated by PyClone. CITUP uses information from multiple samples and can infer clonal populations and their frequencies while satisfying phylogenetic constraints. To ensure accurate tree construction, clusters containing only one mutation were excluded from the input to CITUP if the mutation's role is unclear from previous literature. This filter removed 42 mutations from a total of 1577 mutations, representing less than 2.7% of all clustered mutations. In addition, in GB-A2, clusters which limited to only one organ were excluded from the analysis because the calculation for the phylogenetic tree using CITUP took more than 1 month when the number of clusters was ≥14. Analyzed results were visualized using MapScape for multi-regional specimens and TimeScape for longitudinal or putative longitudinal datasets, as appropriate (*Smith et al., 2017*). In all analysis steps, the data were adjusted for the tumor purity values of each tumor.

Mutational signature analysis was conducted using Mutalisk (*Lee et al., 2018*) and validated with Signal (*Degasperi et al., 2020*) and MuSiCa (*Díaz-Gay et al., 2018*). The COSMIC Mutational Signatures v2 (*Alexandrov et al., 2013*) was used as a reference. To compare with the other GBAC cohorts, we additionally analyzed two public datasets of GBAC from the MSKCC and the Shanghai group (*Narayan et al., 2019*; *Li et al., 2014*), which could be downloaded from the cBioPortal (https://www.cbioportal.org/, accessed on February 1, 2022; *Cerami et al., 2012*).

### *HER2* IHC and SISH

The histologic sections from individual FFPE tissues were deparaffinized and dehydrated. IHC and SISH analysis of *HER2*-positive cells was conducted by the board-certified pathologist using PATHWAY anti-*HER2*/neu antibody (4B5; rabbit monoclonal; Ventana Medical Systems, Tucson, AZ, USA) and a staining device (BenchMark XT, Ventana Medical Systems, Tuscon, AZ, USA), respectively, as previously described (*Koh et al., 2019*). Signals from 20 tumor cells were counted and a *HER2*/CEP17 ratio ≥2.0 was defined as *HER2* amplification (*Wolff et al., 2018*). Wilcoxon rank-sum test is used to compare the mean *HER2*/CEP17 ratio of BilIN versus primary tumor.

## Acknowledgements

This study was supported by a grant from Seoul National University Bundang Hospital Research Fund (No. 16-2021-001) and the Small Grant for Exploratory Research (SGER) program (NRF-2018R1D1A1A02086240) of the National Research Foundation (NRF), Korea. The authors would like to express the deepest respect to the two patients who donated their bodies for this study after death.

## Additional information

### Funding

| Funder | Grant reference number | Author |
|---|---|---|
| Seoul National University Bundang Hospital Research Fund | No. 16-2021-001 | Ji-Won Kim |
| Small Grant for Exploratory Reserach | NRF-2018R1D1A1A02086240 | Ji-Won Kim |

The funders had no role in study design, data collection and interpretation, or the decision to submit the work for publication.

### Author contributions

Minsu Kang, Data curation, Software, Formal analysis, Validation, Visualization, Methodology, Writing – original draft, Writing – review and editing; Hee Young Na, Resources, Data curation, Formal analysis, Validation, Visualization, Methodology, Writing – original draft, Writing – review and editing; Soomin Ahn, Conceptualization, Resources, Formal analysis, Supervision, Validation, Visualization, Methodology, Writing – original draft, Project administration, Writing – review and editing; Ji-Won Kim, Conceptualization, Resources, Data curation, Software, Formal analysis, Supervision, Funding acquisition, Validation, Visualization, Methodology, Writing – original draft, Project administration, Writing – review and editing; Sejoon Lee, Soyeon Ahn, Ju Hyun Lee, Jeonghwan Youk, Haesook T Kim, Formal analysis, Methodology, Writing – original draft, Writing – review and editing; Kui-Jin Kim, Resources, Methodology, Writing – original draft, Project administration, Writing – review and editing; Koung Jin Suh, Jun Suh Lee, Se Hyun Kim, Jin Won Kim, Yu Jung Kim, Keun-Wook Lee, Yoo-Seok Yoon, Jee Hyun Kim, Jin-Haeng Chung, Ho-Seong Han, Jong Seok Lee, Resources, Writing – original draft, Project administration, Writing – review and editing

### Author ORCIDs

Minsu Kang ![ORCID] http://orcid.org/0000-0001-6491-2277
Soomin Ahn ![ORCID] http://orcid.org/0000-0002-1979-4010

Ji-Won Kim http://orcid.org/0000-0001-6426-9074
Se Hyun Kim http://orcid.org/0000-0002-2292-906X

### Ethics

Human subjects: This study was conducted according to the guidelines of the Declaration of Helsinki and approved by the Institutional Review Board of Seoul National University Bundang Hospital (Number: B-1902/522-303). Informed consent was waived because of the retrospective and anonymous nature of the study.

### Decision letter and Author response

Decision letter https://doi.org/10.7554/eLife.78636.sa1
Author response https://doi.org/10.7554/eLife.78636.sa2

## Additional files

### Supplementary files

• Supplementary file 1. Tables. (A) Baseline characteristics of samples are summarized. (B) Mutational signatures of our dataset are analyzed by three different tools, Mutalisk, Signal, and MuSiCa.

• Supplementary file 2. Detected somatic alternations in 11 patients with gallbladder adenocarcinoma (GBAC). Full list of mutations called in our cohort.

• MDAR checklist

### Data availability

The raw sequence data underlying this manuscript are available as fastq files at the NCBI SRA database under the BioProject number PRJNA821382.

The following dataset was generated:

| Author(s) | Year | Dataset title | Dataset URL | Database and Identifier |
|---|---|---|---|---|
| Kang M, Na HY, Ahn S, Kim J-W, Lee S, Ahn S, Lee JH, Youk J, Kim HT, Kim K-J, Suh KJ, Kim SH, Kim JW, Kim YJ, Lee KW, Yoon YS, Kim JH, Lee JS, Ho-Seong ZH, Jin-Haeng C | 2022 | Whole-exome sequencing of gallbladder carcinoma | https://www.ncbi.nlm.nih.gov/sra/PRJNA821382 | NCBI Sequence Read Archive, PRJNA821382 |

The following previously published datasets were used:

| Author(s) | Year | Dataset title | Dataset URL | Database and Identifier |
|---|---|---|---|---|
| Narayan RR | 2018 | Gallbladder Cancer (MSK, Cancer 2018) | https://www.cbioportal.org/study?id=gbc_msk_2018 | cBioPortal, Gallbladder-Cancer-(MSK) |
| Li M | 2014 | Gallbladder Carcinoma (Shanghai, Nat Genet 2014) | https://www.cbioportal.org/study?id=gbc_shanghai_2014 | cBioPortal, Gallbladder-Carcinoma-(Shanghai) |

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
