## [Editor Report]

This is the first dedicated study of clonal evolution in gallbladder cancer that involves precancerous, transformed primary and metastatic lesions. The main insights include the finding of subclonal diversity in precancerous lesions, a degree of bottlenecking during transformation but maintenance of some clonal complexity through metastases.

---

## [Decision Letter]

**Decision letter after peer review:**

Thank you for submitting your article "Gallbladder adenocarcinomas undergo subclonal diversification and selection from precancerous lesions to metastatic tumors" for consideration by *eLife*. Your article has been reviewed by 2 peer reviewers, and the evaluation has been overseen by a Reviewing Editor and W Kimryn Rathmell as the Senior Editor. The following individual involved in the review of your submission has agreed to reveal their identity: Srivatsan Raghavan (Reviewer #1).

Essential revisions:

Please answer reviewers' comments. Overall:

1. Please give inclusion, exclusion, and sample selection criteria indicating which are fresh frozen, which are FFPE and how many samples and sample types are included for each patient. Also samples sites and types of samples. (Surgical vs biopsy).

2. Please clarify the presentation of data and figures.

3. Please explain how you accounted for intratumour heterogeneity.

4. Please describe reconstructing clonal phylogenies and how you dealt with multiple samples per patient.

5. Please analyse FFPE samples with the appropriate adjustments for noise relative to fresh frozen.

6. Please discuss the seeming variation in results of mutational signature analysis as quoted in the abstract versus those in Figure 4.

7. Please re-consider the inference of met-met spread and whether it is really supported by the data.

*Reviewer #1 (Recommendations for the authors):*

1. In Figure 1A, it would be helpful to indicate the site of disease associated with each tissue sample, and also to indicate which specimens were biopsies versus resection tissue.

2. As discussed above in the public review, the authors show variation across different sites of disease and across time, but it is somewhat difficult to distinguish between these in the figures. It would be helpful if Figures 2 and 3 could more clearly distinguish between spatial and temporal variation in clonality, particularly in terms of tracing and more effectively visualizing clonal variation across space versus over time. In addition, the authors don't comment on genomic variation in relation to intervening therapy, and this would be helpful particularly if any notable patterns were observed that may have clinical implications.

3. P. 6, lines 7-11 – In patients with concurrent BilIN and GBAC in the same gall bladder resection specimen, how were these identified? Were these lesions observed by a pathologist in different locations and each isolated separately? Additional details in the methods section would be helpful in this regard.

4. P.8, lines 16-20 – "The metastatic lesions were uniformly polyclonal. In GB-A1, GB-A2, and GB-S4, which contained two or more distant metastatic lesions, the clonal compositions of metastatic lesions were heterogeneous. However, metastatic lesions in one organ or adjacent organs showed similar clonal compositions. In GB-A1 (Figure 2A), abdominal wall 1-4 did not contain clone C, and liver 1-3 did not contain clone F." These statements seem contradictory, particularly the statement referring to "similar clonal compositions" and the following one which provides a counterexample. The conclusion that metastatic lesions in nearby organs are similar is only supported by a few cases and it is unclear that the authors are really powered to state this definitively.

---

## [Author Response]

Essential revisions:Please answer reviewers' comments. Overall:1. Please give inclusion, exclusion, and sample selection criteria indicating which are fresh frozen, which are FFPE and how many samples and sample types are included for each patient. Also samples sites and types of samples. (Surgical vs biopsy).

Thank you for the important comment.

1) We have added the inclusion criteria to the Methods section (Page 15, Line 3-7). There were no specific exclusion criteria and all the obtained samples from the participants were analyzed.

2) A total of 15 samples obtained by rapid autopsy were fresh-frozen, and 43 samples obtained by surgery or biopsy were formalin-fixed paraffin-embedded (FFPE) specimens. We mentioned this in the Results section (Page 5, Line 8-10).

3) We revised Supplementary File 1A. The number of samples, methods for sample acquisition (autopsy vs. surgery vs. biopsy), and type of sample (fresh-frozen vs. FFPE) per each patient are now shown in Supplementary File 1A.

2. Please clarify the presentation of data and figures.

Thank you for your advice. Throughout the revision process, meaningful updates were made, and as a result, the presentation of the data and figures has been made clearer. The important changes are summarized below.

– In Figure 1A, the anatomical information of the obtained samples has been indicated.

– In Figure 1B, the distinction between the tumor suppressor gene and oncogene was removed.

– In Figures 2 and 3, the time point at which the tissue was obtained and the anatomical location of the tissue are clearly separated.

– Figure 4B was moved to Figure 4—figure supplement 2.

3. Please explain how you accounted for intratumour heterogeneity.

Thank you for the important comment. The cellular prevalence of the truncal clone – clone A in every sample – was assumed to be 1.0, and the relative cellular prevalence of the remaining subclones was calculated. The relative proportions of multiple clones were visualized using MapScape and TimeScape (Smith MA et al., Nat Methods. 2017). In Figures 2 and 3, the clonal composition varied from sample to sample, and this is the intratumoral heterogeneity identified in our study.

4. Please describe reconstructing clonal phylogenies and how you dealt with multiple samples per patient.

Thank you for the constructive comment. CITUP (Malikic S et al., Bioinformatics. 2015) infers clonal populations and their frequencies while satisfying phylogenetic constraints and can exploit data from multiple samples. In our study, phylogenetic trees were constructed using the CITUP (0.1.0) command, ‘run_citup_qip.py’. We mentioned this in the Methods section (Page 16, Line 23 – Page 17, Line 1).

5. Please analyse FFPE samples with the appropriate adjustments for noise relative to fresh frozen.

We appreciate your valuable comment. According to previous literature (Chen G et al., Mol Diagn Ther. 2014, Moore DA et al., ESMO Open. 2019), sequencing artifacts induced by formalin fixation satisfy the following conditions: (1) sample specific mutations, (2) C:G > T:A variants, and (3) VAF < 5%. We summarized the mutations satisfying the above conditions in Author response table 1 (for review only). However, we did not remove these mutations in the analysis of the FFPE samples due to the following reasons: (1) it was impossible to determine which mutation is actual noise by formalin fixation, and (2) sample-specific subclones still existed with little influence on the phylogenetic tree structure even after all the potential artifacts have been eliminated.

**Author response table 1. sa2table1:** Proportion of mutations satisfying the conditions: (1) C:G > T:A variants, and (2) VAF < 5% in sample-specific clusters.

Patient ID	Sample	Type of sample	Sample specific clusters^*^	Number of mutations in sample-specific clusters^†^	Number of filtered mutations in sample-specific clusters	Percentage (%)
GB-A1	BilIN	FFPE	2	34	3	8.8
	CBD	FFPE	3	121	13	10.7
GB-S1	BilIN	FFPE	2	9	3	33.3
	GB	FFPE	4	18	7	38.9
	Distant LN	FFPE	7	6	0	0.0
GB-S2	BilIN	FFPE	6	8	3	37.5
	GB	FFPE	0	7	2	28.6
GB-S3	BilIN	FFPE	3	7	1	14.3
	GB	FFPE	2	3	0	0.0
GB-A2	GB	Fresh-frozen	2	40	0	0.0
	Lung (old)	FFPE	10	17	1	5.9
	Lung	Fresh-frozen	5	15	0	0.0
	Chest wall	Fresh-frozen	3	23	0	0.0
	Liver	Fresh-frozen	4	31	0	0.0
	Mesentery	Fresh-frozen	6	8	2	25.0
GB-S4	Colon wall	FFPE	3	36	0	0.0
	Distant LN	FFPE	1	9	2	22.2
GB-S5	GB	FFPE	3	13	0	0.0
GB-S6	GB	FFPE	1	5	0	0.0
	Lung	FFPE	2	7	0	0.0
GB-S7	GB	FFPE	0	11	0	0.0
	Regional LN	FFPE	3	5	0	0.0
GB-S8	GB	FFPE	2	35	4	11.4
	Regional LN	FFPE	0	81	3	3.7

In FFPE samples, sample-specific mutations were found in 11 of 31 samples (35.5%). The number of possible artifacts satisfying the conditions was median 1/sample (range, 0–13/sample), and their proportion among sample-specific mutations was median 5.9% (range, 0–38.9%). In fresh frozen samples, sample-specific mutations were identified in 1 of 14 samples (7.1%). The number of mutations satisfying the conditions was 2, and their proportion among sample-specific mutations was 25%. Although the number of mutations satisfying the artifact conditions was higher in the FFPE samples than in the fresh frozen samples, there is no definitive way to determine which of these is real noise induced by formalin fixation. In addition, there was no sample-specific subclone consisting merely of mutations satisfying the above conditions.

BilIN, biliary intraepithelial neoplasia, GB, gallbladder; LN, lymph node; CBD, common bile duct; FFPE, formalin-fixed paraffin-embedded.

^*^See Figure 1—figure supplement 1.

^†^See Supplementary File 2.

References for Essential Revisions #5

Chen G, Mosier S, Gocke CD, Lin MT, Eshleman JR. Cytosine deamination is a major cause of baseline noise in next-generation sequencing. Mol Diagn Ther. 2014;18(5):587-93.

Moore DA, Kushnir M, Mak G, Winter H, Curiel T, Voskoboynik M, et al. Prospective analysis of 895 patients on a UK Genomics Review Board. ESMO Open. 2019;4(2):e000469.

6. Please discuss the seeming variation in results of mutational signature analysis as quoted in the abstract versus those in Figure 4.

Thank you for the important comment. The sentences about mutational signatures in the abstract were a summary of Figure 4C and had been properly described. If Figure 4B confused you, we sincerely apologize to you for any possible misunderstandings. As stated in the manuscript, Figure 4B denotes mutational signature analysis according to the 4 types of samples (BilIN, GB, regional LN metastasis, and distant metastasis) while Figure 4C indicates the analyzed results according to the timing of development during clonal evolution (early carcinogenesis, late carcinogenesis, and metastasis). We suggest that the criterion of Figure 4C makes more sense considering the evolutionary trajectories in cancers. To avoid confusion, we moved Figure 4B to Figure 4—figure supplement 2 and revised the phrase "developmental stages of cancer" to "type of sample" in this figure.

7. Please re-consider the inference of met-met spread and whether it is really supported by the data.

Thank you for the important comment. We agree with reviewer #1's comment 3 that the rationale for metastasis-to-metastasis spread for GB-A1 is weak. Hence, this part was removed from the manuscript.

However, our data still suggests that the possibility of metastasis-to-metastasis spread is shown in GB-A2. In GB-A2, clone F, which is not present in GB, is dominant in the liver and mesentery, and this finding can be evidence of metastasis-to-metastasis. In addition, the clonal composition of GB is different from those of the lung and chest wall, while the clonal composition of the lung and chest wall is similar.

For this part, Figure 3A and its description in the manuscript have been updated (Page 12, Line 9-12).

Reviewer #1 (Recommendations for the authors):1. In Figure 1A, it would be helpful to indicate the site of disease associated with each tissue sample, and also to indicate which specimens were biopsies versus resection tissue.

Thank you for the important comment. Reflecting on your comment, the anatomical information of the obtained samples has been updated in Figure 1A. As you can check by combining the color of the box (Surgery or Autopsy) and '' mark in Figure 1A, all were resection samples from surgery or autopsy except for the liver tissue of GB-S5 (biopsy). In addition, information on whether the tissue was obtained by resection or biopsy was added to Supplementary File 1A.

2. As discussed above in the public review, the authors show variation across different sites of disease and across time, but it is somewhat difficult to distinguish between these in the figures. It would be helpful if Figures 2 and 3 could more clearly distinguish between spatial and temporal variation in clonality, particularly in terms of tracing and more effectively visualizing clonal variation across space versus over time. In addition, the authors don't comment on genomic variation in relation to intervening therapy, and this would be helpful particularly if any notable patterns were observed that may have clinical implications.

Thank you for the important comment about (1) figures and (2) genomic variation about intervening therapy.

1) We have modified Figures 2 and 3 to distinguish between spatial and temporal variation more clearly.

2) Although it would have been better to compare genetic variation before and after treatment, especially when targeted therapies are used, only 2 out of 11 patients received targeted agents in our cohort. The GB-A1 patient was treated with varlitinib, both an *EGFR* and *HER2* inhibitor, and then ceralasertib, which targets the DNA damage repair pathway. There was no difference in *HER2* copy number before and after using varlitinib (Figure 2A). The number of mutations before the administration of ceralasertib was a median of 49 (range, 44–52), and the number of mutations after administration was a median of 60 (range, 48–104). However, it is difficult to conclude that the increased number of mutations was caused by ceralasertib because the administration period was as short as 2 months and the number of patients was only one. The GB-A2 patient was treated with a RON receptor tyrosine kinase inhibitor. After treatment, clone F containing *PRKCD* p.I153L expanded. However, it is difficult to give meaning to a single clinical case. Nevertheless, we agree with your opinion that we can get more clinical insights from sequencing multiple specimens before and after treatment.

3. P. 6, lines 7-11 – In patients with concurrent BilIN and GBAC in the same gall bladder resection specimen, how were these identified? Were these lesions observed by a pathologist in different locations and each isolated separately? Additional details in the methods section would be helpful in this regard.

Thank you for the constructive comment. Our board-certified pathologists (Prof. Soomin Ahn and Hee Young Na) identified and dissected BilIN and GBAC lesions from tissue slides. We added this information to the Method section (Page 15, Line 8-9).

4. P.8, lines 16-20 – "The metastatic lesions were uniformly polyclonal. In GB-A1, GB-A2, and GB-S4, which contained two or more distant metastatic lesions, the clonal compositions of metastatic lesions were heterogeneous. However, metastatic lesions in one organ or adjacent organs showed similar clonal compositions. In GB-A1 (Figure 2A), abdominal wall 1-4 did not contain clone C, and liver 1-3 did not contain clone F." These statements seem contradictory, particularly the statement referring to "similar clonal compositions" and the following one which provides a counterexample. The conclusion that metastatic lesions in nearby organs are similar is only supported by a few cases and it is unclear that the authors are really powered to state this definitively.

We agree with your important comment. The contradictory and unclear sentences were corrected to convey the meaning accurately, and the overall statement was softened:

"The metastatic lesions were uniformly polyclonal. In GB-A1, GB-A2, and GB-S4, which contained two or more distant metastatic lesions, the clonal composition of tissues obtained from the same or adjacent organs showed a similar tendency, while the clonal composition of anatomically distant organs was distinct from each other." (Page 8, Line 13-16)